

# Simulating the effects of weather and climate on large wildfires in France

Renaud Barbero[1], Thomas Curt[1], Anne Ganteaume[1], Eric Maillé[1], Marielle Jappiot[1], and Adeline Bellet[1]

[1]Irstea, Mediterranean Ecosystems and Risks, Aix-en-Provence, France

*Correspondence to:* R. Barbero (renaud.barbero@irstea.fr)

**Abstract.** Large wildfires across parts of France can cause devastating damages which put lives, infrastructures, and natural ecosystem at risk. One of the most challenging questions in the climate change context is how these large wildfires relate to weather and climate and how they might change in a warmer world. Such projections rely on the development of a robust modeling framework linking wildfires to atmospheric variability. Drawing from a MODIS product and a gridded meteorological
dataset, we derived a suite of biophysical and fire danger indices and developed generalized linear models simulating the probability of large wildfires (>100 ha) at 8-km spatial and daily temporal resolutions across the entire country over the MODIS period. The models were skillful in reproducing the main spatio-temporal patterns of large wildfires across different environmental regions. Long-term drought was found to be a significant predictor of large wildfires in flammability-limited systems such as the Alpine and Southwest regions. In the Mediterranean, large wildfires were found to be associated with
both short-term fire weather conditions and longer-term soil moisture deficits, collectively facilitating the occurrence of large wildfires. Simulated probabilities during the day of large wildfires were on average 2-3 times higher than normal with respect to the mean seasonal cycle. The model has wide applications, including improving our understanding of the drivers of large wildfires over the historical period and providing a basis to estimate future changes to large wildfire from climate scenarios.

**1 Introduction**

Large wildfires in France have received much attention recently due to the threat they pose to ecosystems, society, property and the economy. In the Mediterranean, large wildfires threat many of the ecosystems components (Pausas et al., 2008) and can induce potential shifts in plant composition and structure (Vennetier and Ripert, 2009; Frejaville et al., 2013) or soil losses. Additionally, the growth of the wildland-urban interface (WUI) has increased wildfire risk, the cost of suppression
and our vulnerability across the region (Lampin-Maillet et al., 2011; Modugno et al., 2016; Ruffault and Mouillot, 2017; Fox et al., 2018) and will continue to do so given future demographic trends (Radeloff et al., 2018). Although wildfire extent does not systematically reflect wildfire intensity and the related-impacts (Tedim et al., 2018), large wildfires are usually the most destructive for both ecosystems and infrastructures.





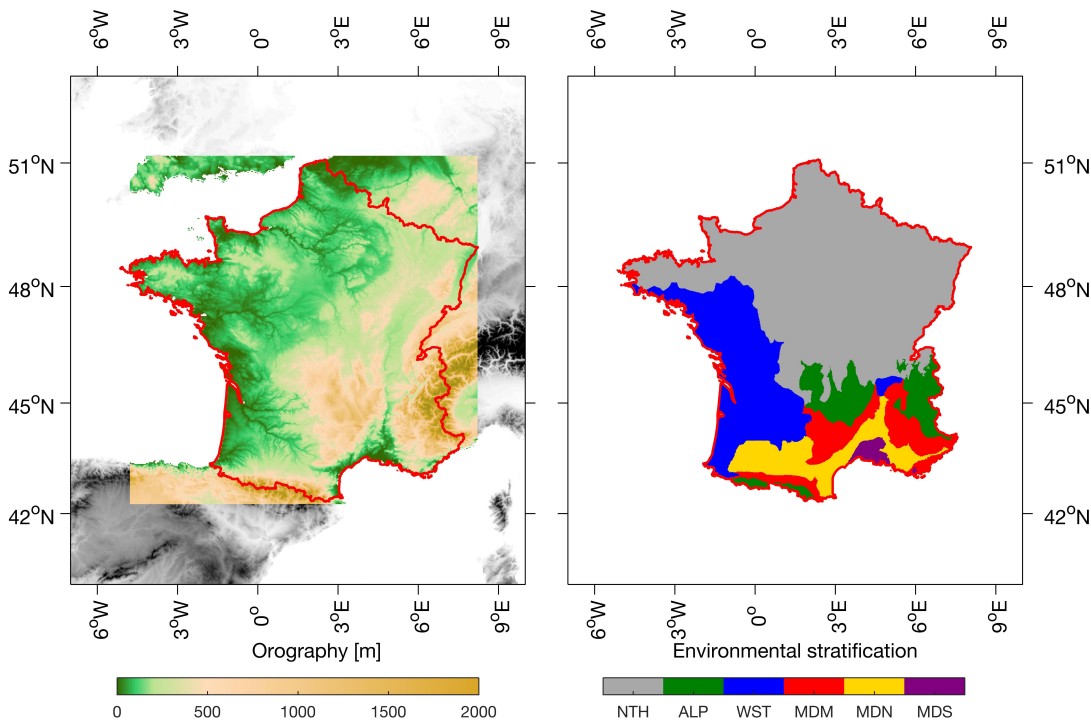

**Figure 1.** Left) Orography in France. The red contour shows the outlines of the investigated French region. Right) Environmental stratification based on climate data, data on the ocean influence and geographical position (Metzger et al., 2005; Jongman et al., 2006). Abbreviations: NTH – North (Atlantic Central in Metzger et al. (2005)); ALP - Alpine; WST - WEST (Lusitanean in Metzger et al. (2005)); MDM - Mediterranean Mountains; MDN - Mediterranean North; MDS - Mediterranean South.

Wildfire ignitions in Europe were strongly related to a range of human activities (Ganteaume et al., 2013) with arson and negligence being the main wildfire ignition causes in the French Mediterranean (Ganteaume and Jappiot, 2013; Curt et al., 2016; Ganteaume and Guerra, 2018). Despite the accidental and unintentional nature of most wildfire ignitions, wildfire spread in the Mediterranean is generally enabled by a range of weather-to-climate processes operating at different timescales such as long-term drought (Hernandez et al., 2015b; Turco et al., 2017) and synchronous favorable large-scale weather conditions including synoptic blocking (Hernandez et al., 2015a) or the Atlantic ridge weather type (Ruffault et al., 2017b). Previous efforts have also distinguished different mechanisms responsible for large wildfire: wind speed and anomalously warm conditions (Ruffault et al., 2016). While wind-induced wildfires may arise due to strong winds that accelerate the rate of spread in a specific direction, heat-induced wildfires (also called plume-driven wildfires) arise due to anomalously warm conditions



that increase fuel dryness and flammability and facilitate wildfire spread in all directions (Lahaye et al., 2017) contingent on topography and fuel structure. Collectively, heat wave, wind speed and drought conditions during previous months have been shown to enhance the potential for large wildfire (Hernandez et al., 2015a, b; Ruffault et al., 2017a). However, most of these previous efforts have exploited regional datasets of burned area across parts of Southeast France commencing in early 1970s

and little attention has been devoted to understanding processes in other regions.

A substantial reduction in wildfire activity was observed in the 1990s across the French Mediterranean due to suppression and prevention strategies (Ruffault and Mouillot, 2015; Curt and Frejaville, 2017), decoupling wildfire trends from climate expectations (Fréjaville and Curt, 2017). However, the 2003 heat wave have induced wildfire prone meteorological conditions across the region impeding suppression efforts and promoting 2003 as one of the most extreme year in terms of burned area

over the last six decades (Ganteaume and Barbero, submitted). The continued intensification and increased frequency of heat waves in the future due to climate change (Vautard et al., 2013; Guerreiro et al., 2018) raises legitimate concerns about the sustainability of current fire policies and strategies. Additionally, the accumulation of fuel loads due to this wildfire suppression effort within a long-term forest recovery context across the Mediterranean (Abadie et al., 2017) is widely thought to have created favorable ground conditions for wildfire spread and the occurrence of large wildfires (Curt and Frejaville, 2017).

In this context, it is essential to develop a modeling framework resolving the complex relationships linking weather-to-climate variability to the occurrence of large wildfires. Such model is still lacking due to observational inhomogeneities in wildfire detection across the country, hampering the compilation of a homogeneous database. Drawing from a global remote sensing database of wildfire, we sought here to develop a nation-wide statistical model including wildfire-prone regions over-looked in previous studies. The model is expected to advance our understanding of processes and drivers of large wildfires and

to provide guidance on how weather and climate variability may increase the occurrence of large wildfires in a warmer world across the whole country.

## 2 Data and methods

### 2.1 Wildfire data

We used the Moderate Resolution Imaging Spectroradiometer (MODIS) Firecci v5.0 product developed within the frame-

work of the European Space Agency's Climate Change Initiative (CCI) programme and available on the period 2001–2016 (Chuvieco et al., 2016). This product is based on MODIS on board of the Terra polar heliosynchronous orbiting satellite. The burned area algorithm combined temporal changes in near-infrared MERIS-corrected reflectances based on MOD09GQ of the MODIS sensor at 250-m spatial resolution with active fire detection from the standard MODIS thermal anomalies product, following a two-phase algorithm (Alonso-Canas and Chuvieco, 2015). Complementary to the surface reflectance product, the

daily MOD09GA Collection 6 product was also used to extract information on the quality of the data. Although small wildfires are generally difficult to detect with satellite observations due to the timing of the scan or cloud-cover impairment of remote sensing, our focus on large wildfires is expected to minimize this uncertainty.





**Table 1.** Candidate variables in the modeling framework.

| Name | Acronym | Category |
|------|---------|----------|
| 1. Mean temperature | TMEAN | Meteorological Variable |
| 2. Minimum temperature | TMIN | Meteorological Variable |
| 3. Maximum temperature | TMAX | Meteorological Variable |
| 4. Relative humidity | RH | Meteorological Variable |
| 5. Wind speed | WS | Meteorological Variable |
| 6. Precipitation | PRCP | Meteorological Variable |
| 7. Fine Fuel Moisture Code | FFMC | Fire-Weather metric |
| 8. Duff Moisture Code | DMC | Fire-Weather metric |
| 9. Drought Code | DC | Fire-Weather metric |
| 10. Initial Spread Index | ISI | Fire-Weather metric |
| 11. Build-Up Index | BUI | Fire-Weather metric |
| 12. Fire Weather Index | FWI | Fire-Weather metric |
| 13. Forest McArthur Fire Danger Index | FFDI | Fire-Weather metric |
| 14. F-Index | FINDEX | Fire-Weather metric |
| 15. Nesterov Fire Danger Index | NFDI | Fire-Weather metric |
| 16. Fosberg Fire Weather Index | FFWI | Fire-Weather metric |
| 17. Effective drought Index | EDI | Drought metric |
| 18. Potential Evapotranspiration | PET | Drought metric |
| 19. Standardized Precipitation Index | SPI | Drought metric |
| 20. Keetch Byram Drought Index | KBDI | Drought metric |
| 21. Soil Wetness Index | SWI | Soil Moisture metric |

**Table 2.** Contribution of large wildfires (>100 ha) to (first column) regional total burned area, (second column) national total burned area and (third column) pearson correlations between the frequency of large wildfires and regional total burned area. The symbol * indicates significant correlation at the 95% confidence level.

| Env. Region | Contribution to regional BA | Contribution to national BA | $r_{(LWF,TBA)}$ |
|-------------|------------------------------|------------------------------|-----------------|
| North | 61.8% | 1.9% | 0.80* |
| Alpine | 69.1% | 2.3% | 0.88* |
| West | 63.6% | 5.4% | 0.96* |
| Mediterranean Mountains | 83.9% | 34.8% | 0.99* |
| Mediterranean North | 75.9% | 25.7% | 0.97* |
| Mediterranean South | 72.4% | 7.1% | 0.86* |





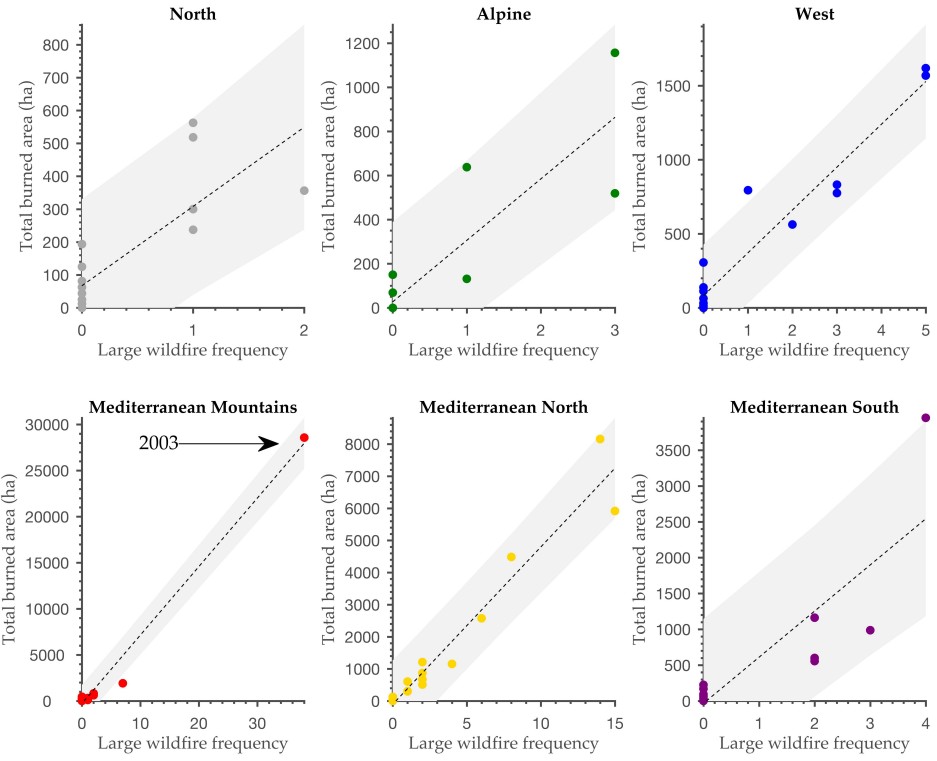

**Figure 2.** Interannual relationships between the annual frequency of large wildfires (>100 ha) and the total annual burned area. The Linear fitting and the 95% confidence intervals are shown for each region.

We excluded MODIS fires located within agricultural lands using CORINE Land Cover 2012 data (https://land.copernicus.eu/pan-european/corine-land-cover) as well as prescribed fires related to pastoral practices during the cool season (November-March) since these fires are generally under control and do not put infrastructures or ecosystems at risk. MODIS pixels spatially and temporally adjacent were aggregated using the location and the date of the first detection to form consistent wildfire events.

5  Pixels belonging to the same wildfire event were required to be within a maximum distance of 4 pixels (to minimize inaccuracies in burned area detection within a pixel) and to have adjacent burning dates. The 22,785 MODIS pixels extracted from 2001-2016 across France were found to form 894 distinct wildfire events. We then defined large wildfires as wildfires whose size exceeds 100 ha (N=156 large wildfires) following Ganteaume and Barbero (submitted), a threshold corresponding here to the $83^{th}$ percentile of the distribution of wildfire extent. The average large wildfires extent was found to be 398 ha,



with the largest wildfire reaching 7675 ha. Finally, we regridded this information onto an 8-km resolution grid to facilitate the comparison with meteorological data (see section 2.2).

## 2.2 Weather and climate data

Meteorological variables were obtained from the quality-controlled SAFRAN dataset providing minimum and maximum tem-
perature, relative humidity, precipitation and wind speed over France from 2001-2016 (Vidal et al., 2009, 2010, 2012) on a daily basis and over an 8-km grid.

Drawing from the SAFRAN meteorological data, we derived a suite of fire weather and drought variables (see Table 1) intended to reflect different timescales of variability that are widely thought to facilitate wildfire spread from synoptic (weather) to interannual (climate) scales (Barbero et al., 2015a; Nogueira et al., 2017). Fire weather variables included the Canadian
Forest Fire Weather Index system (Fine Fuel Moisture Code, Build-Up Index, Duff Moisture Code, Initial Spread Index, Drought Code, Fire Weather Index) (Van Wagner, 1987). Although these indices were empirically calibrated for estimating whether atmospheric conditions and fuel moisture content are prone to wildfire development in Canada (Van Wagner, 1987), the FWI system has proven useful in Mediterranean regions (Dimitrakopoulos et al., 2011; Fox et al., 2018; Lahaye et al., 2017) as well as in Alpine environments (Dupire et al., 2017). We also included in our analysis other fire weather indices
that have been shown useful in estimating fire danger conditions across parts of the world including the Forest McArthur Fire Danger Index (Dowdy et al., 2010), the F-Index (Sharples et al., 2009), the Nesterov Fire Danger Index (Nesterov, 1949) and the Fosberg Fire Weather Index (Fosberg, 1978). Further information on each of these fire weather variables and how they relate to wildfire activity can be found in the literature.

Additionally, we used a series of fast- and slow-reacting drought indices to detect flash and chronic drought that are often
associated with large wildfires. These indices include potential evapotranspiration based on the Penman–Monteith equation, the Keetch-Byram Drought Index (Keetch and Byram, 1968), the Effective Drought Index (Byun and Wilhite, 1999) integrating here precipitation over the last 30 days to detect short-term precipitation deficit, the Standardized Precipitation Index based on a nonparametric framework (Farahmand and AghaKouchak, 2015) and computed on 6-month windows to detect long-term precipitation deficit and the more sophisticated Soil Wetness Index developed by CNRM (Centre National de la Recherche
Météorologique). This last index was derived from ISBA (Noilhan and Mahfouf, 1996), a soil-biosphere-atmosphere interaction model based on soil characteristics across France. The Soil Wetness Index integrates the propagation of moisture from the superficial surface layer to the root zone (Barbu et al., 2011).

A composite analysis of each of these variables over a period spanning 30 days before to 30 days after the occurrence of large wildfires is provided in the supplementary information (Figures A1 and A2). These figures show the potential and the
relevance of each of these variables to predict large wildfires as shown in previous efforts (Ruffault et al., 2016).

## 2.3 Environmental stratification

The relationships between weather-to-climate and wildfire activity across parts of France is mediated through vegetation, the complex topography of the region (Figure 1, left) alongside human factors (Fréjaville and Curt, 2015; Ganteaume and Long-





Fournel, 2015; Ganteaume and Guerra, 2018). Given the compounding influence of these environmental factors, we developed separate models using an environmental stratification (Figure 1, right) based on climate data, topography and geographical position (Metzger et al., 2005; Jongman et al., 2006), assuming that the weather-to-climate forcing on large wildfires is relatively consistent within each of these regions. The North region (Atlantic Central in (Metzger et al., 2005)), the less fire-prone region, corresponds to a temperate climate where average summer temperatures are relatively low. The Alpine region spans high mountains conditions typical of the alpine ranges of southern Europe which is dominated by conifer forests at high elevation and broadleaves at low elevation. The West region (Lusitanean in (Metzger et al., 2005)), corresponds to the southern Atlantic climate with warm and dry summers and mild and humid winters. Further south, the Mediterranean area is stratified into three distinct environmental regions: the Mediterranean Mountains (hereafter referred to as Mnts in the figures), that combine the influence of both Mediterranean and mountain climates (including various species such as Fagus sylvatica, Pinus sp., Quercus pubescens), the Mediterranean North, which is a holm oak, cork oak dominated-vegetation (Quercus ilex, Quercus suber) and the Mediterranean South, a low elevation area (Figure 1, left) spanning the Rhone delta. The spatial extent of each region allows the pooling of a decent number of MODIS wildfires needed to develop robust models. We however acknowledge the existence of sub-regional variations in human factors (e.g., ignition and suppression) and that other biogeographic units with homogeneous attributes with respect to wildfire regime and climatic conditions may yield different results (Fréjaville and Curt, 2015).

### 2.4 The modeling framework

#### 2.4.1 Generazlized Linear Models

Empirical models linking weather and climate to wildfires have received much attention in the climate change context (Riley and Thompson, 2016) and multiple model specifications have been introduced in the literature (Boulanger et al., 2018) to simulate wildfire activity. We sought here to develop separate models in each environmental region to simulate the probability of large wildfire (given an ignition) at 8-km spatial and daily resolutions based purely on the weather-to-climate forcing. Simulating the day-to-day variability has the advantage of detecting short-duration synoptic conditions otherwise masked in monthly or seasonal timescales.

We used Generalized Linear Models (GLM) with a stepwise regression using all predictors listed in Table 1. GLMs have already been used to simulate the occurrences of large wildfires in other regions of the world (Stavros et al., 2014a, b; Barbero et al., 2014, 2015b) given their ability to model the relationship between a dichotomous variable (presence/absence of large wildfires) and a set of predictor variables. For each day of the MODIS period (2001-2016) and each cell of the 8-km grid, the binomial predictand ($y$) was coded as 1 if a large wildfire was observed, and 0 otherwise. This binary response is modeled as the probability ($P$) to observe a large wildfire via a logistic model with a logit link such as:

$$P(y = 1|x) = \frac{exp(\beta' x)}{1 + exp(\beta' x)}$$





**Table 3.** Equations describing daily large wildfire (>100 ha) probabilities at 8-km for each environmental region. The second column indicates the number of large wildfires observed from 2001-2016. The third column gives the $\beta' x$ parameters and the last column indicates the model selection frequencies, i.e. the percent of bootstraps in which there was agreement.

| Env. Region | # Large wildfires | $\frac{exp(\beta' x)}{1+exp(\beta' x)}$ | Bootstrap $\pi_i$ |
|---|---|---|---|
| North | 6 | $\beta' x = -20.674 + FFMC \times 0.0767$ | 100 |
| Alpine | 7 | $\beta' x = -14.828 + SPI \times (-1.9868)$ | 98 |
| West | 19 | $\beta' x = -16.242 + DC \times 0.0054$ | 100 |
| Mediterranean Mountains | 45 | $\beta' x = -7.4715 + BUI \times 0.00076 + SWI \times (-12.363)$ | 27 |
| Mediterranean North | 61 | $\beta' x = -8.7438 + FWI \times 0.067 + SWI \times (-13.036)$ | 92 |
| Mediterranean South | 12 | $\beta' x = -11.932 + DMC \times 0.0183$ | 100 |

where $\beta' = (\beta_0, \beta_1, ... \beta_p)$ is a vector of coefficients relating probability of wildfires to $p$ covariates via the relationship $\beta' x = \beta_0 + \beta_1 x_1 + ... + \beta_p x_p$. $P$ is intrinsically bounded in the interval [0,1]. We considered each observation of the predictor variables as independent samples despite the inherent spatial autocorrelation and serial correlation. This violates the assumption of independence between samples and overestimates the number of degrees of freedom. However, these effects are mitigated with

the use of a random sampling design in model development (see below), although it is not intended to completely remove the true spatial autocorrelation. Predictor variables that did not significantly improved the model were discarded from stepwise model selection procedure using the Bayesian Information Criterion (BIC) since the penalty for additional parameters is higher in BIC than in Akaike Information Criterion (AIC), consequently favoring more parsimonious models (Murtaugh, 2009). Also, we did not allow interactive and non-linear terms in logistic equations.

**2.4.2   Model selection uncertainty**

While a model may be developed using all 8-km grid cells available through the period and the region, numerous caveats arise that limit the model robustness, particularly given the huge imbalance between 0 (absence) and 1 (presence). Selecting the "best" approximating model from one single sample would raise the following question: would the same model be selected with another sample? The model selection uncertainty is of primary importance (Burnham and Anderson, 2002), especially when

competing models exist to describe the unknown state of the complex climate-wildfires relationship. The use of replications in logistic regressions allows avoiding instability of the results due to sampling bias. We used resampling methods combining the strength of probabilistic and statistical methods (Guns and Vanacker, 2012) to assess model stability and to achieve a proper tradeoff between bias and variance (Burnham and Anderson, 2002). To do so, we conducted a case-control experiment (Keating and Cherry, 2004) and generated 1,000 bootstrap samples to estimate model selection frequencies ($\pi_i$). Each sample includes

all large wildfire occurrences (1) as well as 50,000 randomly chosen non-occurrences (0). Each resampled data corresponds to a specific grid cell during a specific day. Maximum likelihood theory provides estimates of the parameters $\beta'$ and the BIC-best model is found for each bootstrap sample. Finally, the model selection relative frequencies ($\pi_i$) are computed as the sums of the frequencies where model $i$ was selected as best, divided by the total number of bootstraps. We used the model with the





**Table 4.** Spearman rank correlations between large wildfires (>100 ha) observed and that expected from simulated probabilities at the monthly and interannual timescales. The symbol * indicates significant correlation at the 95% confidence level.

| Env. Region | Monthly variations | Interannual variations |
| --- | --- | --- |
| North | 0.56 | 0.45 |
| Alpine | 0.58* | 0.23 |
| West | 0.73* | 0.58* |
| Mediterranean Mountains | 0.68* | 0.59* |
| Mediterranean North | 0.73* | 0.62* |
| Mediterranean South | 0.67* | 0.22 |

highest $\pi_i$ from these bootstrap in subsequent modeling as we considered this model to represent the most stable relationships in a given region. Note that the simulated probabilities were derived using all data available so that the sum of simulated probabilities in a given region reflects the total number of large wildfires.

Receiver-operating characteristic (ROC) plots was used to evaluate the model performance, as ROC statistics provide in-
formation for a range of possible threshold values to classify a grid cell during a specific day as "prone to large wildfire" and gives rapidly an overall idea of model skill. The ROC curve shows the false-positive rate vs the true-positive rate and the area under the ROC curve ranges from 0.5 (random prediction) to 1.0 (perfect prediction). Also, we examined simulated probabilities expressed as anomalies with respect to the mean annual cycle both 80 days prior to and 80 days following observed large wildfire days at the 8-km grid cell level. This allows determining whether simulated probabilities during large wildfires
days were locally higher than what we would expect from the seasonal forcing alone and how fast these large wildfire-prone conditions develop across the season.

## 3 Results and discussion

### 3.1 Model selection and regional variability

In each region, the annual frequency of large wildfires strongly shaped interannual variations in annual burned area (Figure
2), with an overall contribution in total burned area from 2001-2016 ranging from 62% (North) to 84% (Mediterranean Mnts) (Table 2). Mediterranean North and Mediterranean Mountains were the strongest contributors to national total burned area.

Most models were found to be skillful (Figure 3). The AUC was the highest for the Mediterranean North which is also the region with the largest sample of large wildfires, while the Mediterranean South model has the lowest predictive power. In this region, a true positive rate hardly exceeding 0.8 is associated with a false negative rate exceeding 0.7, indicating that large
wildfires in that region are less related to the weather-to-climate forcing.

Table 3 shows the best model selected in each environmental region alongside the relative model selection frequencies ($\pi_i$) from applying BIC to each of the 1,000 bootstrap samples. In five out of the six regions, models were selected as the best in

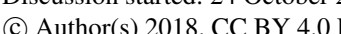



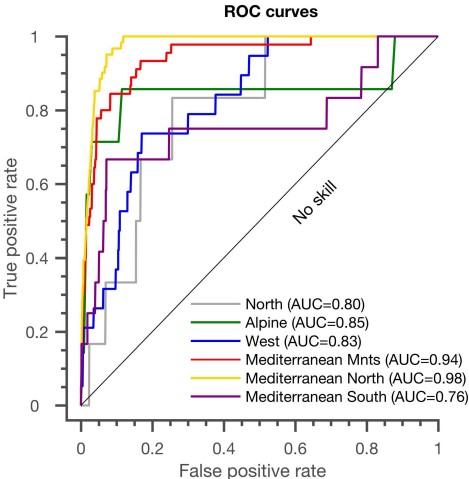

**Figure 3.** Area under the curve (AUC) illustrating the performance of each model.

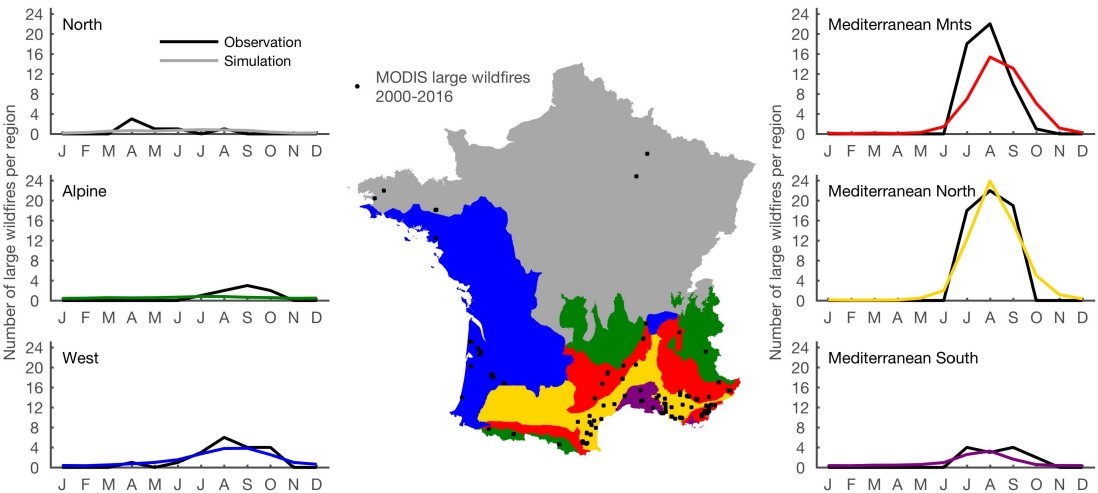

**Figure 4.** Observed (black) and simulated (color) total number of large wildfires (>100 ha) per month in each environmental region. The middle panel shows the location of large wildfires during the April-October season from 2001-2016.



>90% of the simulations. However, the Mediterranean Mountains model has much lower selection frequencies (27%). Note that a lower frequency of selection does not mean that the model is less skillful, but rather that other combinations of predictors are possible. The table also shows the different predictor variables selected in the stepwise procedure and indicates how the weather-to-climate forcing can affect large wildfires very differently depending on what kinds of environmental conditions predominate. In the North, the best model uses only FFMC, an index reflecting the flammability of litter and fine fuels. As opposed to Mediterranean regions where FFMC quickly saturates early summer due to overall low soil moisture conditions, this index seems to be useful in tracking large wildfire potential in more humid climates where fine fuels dominate. By contrast, the Alpine and West best models are based on slow-reacting indices (SPI for the Alpine region and DC for the West region), both reflecting chronic soil moisture deficit and low dead and fuel moisture levels. This suggests that large wildfires in these more humid and more flammability-limited systems are mainly enabled by slow-evolving drought. In both Mediterranean Mountains and Mediterranean North, the best models combine the information provided by the BUI and the FWI (fire weather metrics) with the SWI (soil moisture content metric). In fact, a decrease in the SWI in summer, corresponding to a reduction of plant available soil moisture level, accelerates the vegetation mortality (Barbu et al., 2011), which is widely thought to facilitate wildfire spread. This illustrates how complementary fire weather and soil moisture indices are, and how they may, collectively, improve the ability to track the potential for large wildfire.

It is noteworthy that the commonly accepted effect of wind speed in the Mediterranean region is only revealed through the FWI and not directly through wind speed. Wind speed alone is probably not able to discriminate between large wildfires and non-large wildfires days, this discrimination applies only if other climate conditions are also gathered. Moreover, large wildfires in the Mediterranean are often associated with a cold wind (Ruffault et al., 2017b), with contrasting effects on commonly used fire weather indices that were designed to increase with temperature.

### 3.2 Simulated seasonal and interannual variability

The seasonal cycle, featuring a peak in August in most regions, is well reproduced in the simulations (Figure 4 and Table 4). However, some large wildfires were also seen in the spring and in September in the North and Alpine regions respectively, a feature that is not reproduced in the model due to sample size limitations. Figure 5 shows the spatio-temporal patterns of mean daily simulated probabilities from May-October. The potential for large wildfires emerges in the Mediterranean South first and propagates northwards in the Mediterranean Mountains and along the west coast and then slowly decays in October. An animation of daily simulated probabilities from 2001-2016 is available in the supplementary information (Supp1.mov).

Models were able to simulate, to some extent, interannual variations in large wildfires (Figure 6 and Table 4), including the exceptionnal 2003 outbreak in Mediterranean Mountains. As expected, interannual variance in simulated probabilities was much lower than that of large wildfires observed due to the continuous nature of probabilities (in contrast to the strongly intermittent nature of large wildfire), thereby underestimating (overestimating) the probability of very likely (unlikely) events and resulting in a variance deflation.





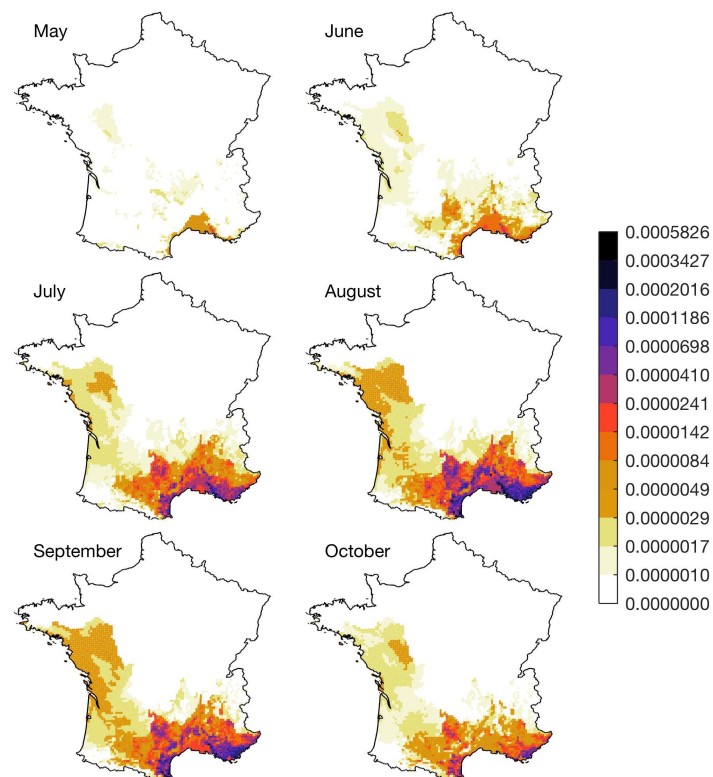

**Figure 5.** Daily large wildfire (>100 ha) probabilities averaged across months from 2001-2016. Note the highly non-linear color bar (probabilities in the highest class are >582 times higher than those in the first class).

### 3.3 Local simulated probabilities

Figure 7 shows the simulated probabilities expressed as anomalies with respect to the mean seasonal cycle during a period spanning 80 days before to 80 days after large wildfires pooled over the entire country. Simulated probabilities progressively increased until the day of large wildfire, reaching values 2-3 times higher than normal, and then slowly decays to normal condi-

5    tions two months later. This temporal pattern was variable across environmental regions (Figure 8) depending on the predictor variables selected (Table 2). In fact, the slowly increasing probabilities evident in the Alpine region and the Mediterranean Mountains align with global change-type drought (Breshears et al., 2005; Ruffault et al., 2017a) while the faster-increasing



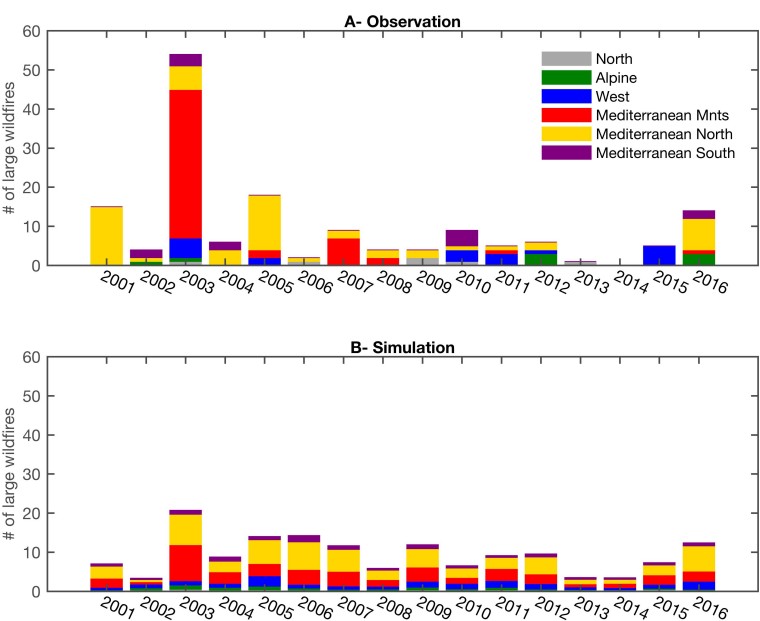

**Figure 6.** Observed (top) and simulated (bottom) total number of large wildfires (>100 ha) per year in each environmental region.

probabilities in the Mediterranean North and South resemble flash drought, highlighting different generating mechanisms across regions.

## 4   Conclusions

This study provides a statistical modeling framework of large wildfires in France from the weather-to-climate forcing. The
5   environmental stratification used here (Metzger et al., 2005) has proven useful to aggregate large wildfires and develop different models and should be considered in future pan-European climate-wildfire modeling efforts. We however acknowledge that this stratification is likely, as any other stratification, to mix-up large wildfires with different causes and different climate drivers. Collectively, metrics from the Canadian FWI system in combination with drought and soil moisture indices were skillful in tracking large wildfire across the country as already demonstrated in regional studies in the French Alps (Dupire et al., 2017)
10   and southeastern France (Ruffault et al., 2017a). In particular, the SWI reflecting the soil moisture available for the plant was useful in predicting large wildfires across the Mediterranean region and seems to complement traditional fire weather indices.



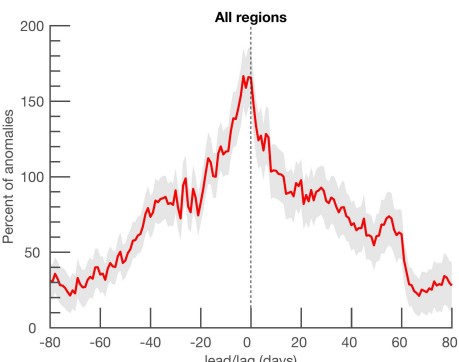

**Figure 7.** Composites of local simulated probabilities (expressed as percent with respect to the mean seasonal cycle, i.e. 100% indicates that probabilities are two times higher than normal) relative to the large wildfire (>100 ha) days. The 95% confidence intervals of the composite means are computed using 1000 bootstrapped datasets. The envelope of confidence indicates the 2.5 and 97.5 percentile of the composite means obtained from the bootstrapped datasets.

Simulated probabilities based on these predictor variables were, on average, 2-3 times higher than normal during the day of large wildfires, highlighting once again the strong control that the weather-to-climate forcing exerts on the occurrence of large wildfires.

Several caveats and limitations apply to our modeling framework. Our model is based purely on weather and climate and

5   ignores human activities (ignition/suppression). Indeed, drought is one component of a complicated wildfire system (Littell, 2018) and our modeling framework is obviously contingent on ignition and fuels. Although human activities adds a less understood and therefore less predictable component (Littell, 2018), including human factors such as population density or WUI (Costafreda-Aumedes et al., 2017) as well as causes of wildfire ignition (Ganteaume and Guerra, 2018) would greatly improve model skill. Future changes to large wildfire regimes over the next several decades depends not only on the climate

10   variability, but also on human ignition, suppression and prevention. Large wildfire occurrences impose multiple challenges to both ecosystems and societies and this vulnerability should also be considered in the more complex contexts of risk assessment.

Finally, our model will serve as a basis to simulate future changes to large wildfires based on future climate projections from the EURO-Cordex project. Such projections will help better understand future changes and will provide the information decision-makers need for successful adaptation to climate change.





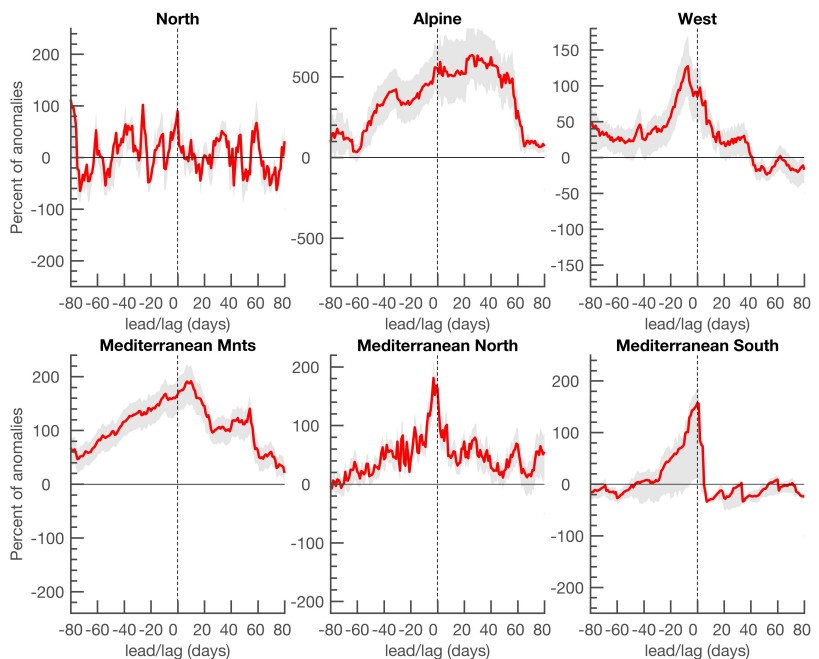

**Figure 8.** Same as figure 7 but for specific regions. Note the different ranges on the y-axis.

**Appendix A**

*Competing interests.* No competing interests.

*Acknowledgements.* TEXT

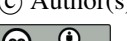

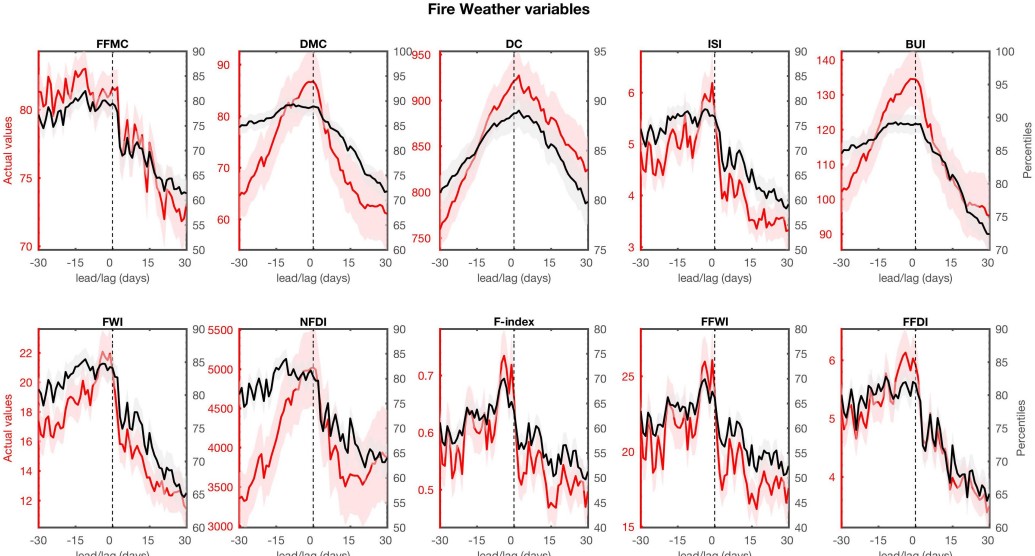

**Figure A1.** Composites of local fire weather variables relative to the large wildfire (>100 ha) days. Actual values are shown in red while percentiles (computed from the local distribution of each 8-km grid cell) are shown in black. The 95% confidence intervals of the composite means are computed using 1000 bootstrapped datasets. The envelope of confidence indicates the 2.5 and 97.5 percentile of the composite means obtained from the bootstrapped datasets. Acronyms are defined in Table 1.

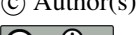



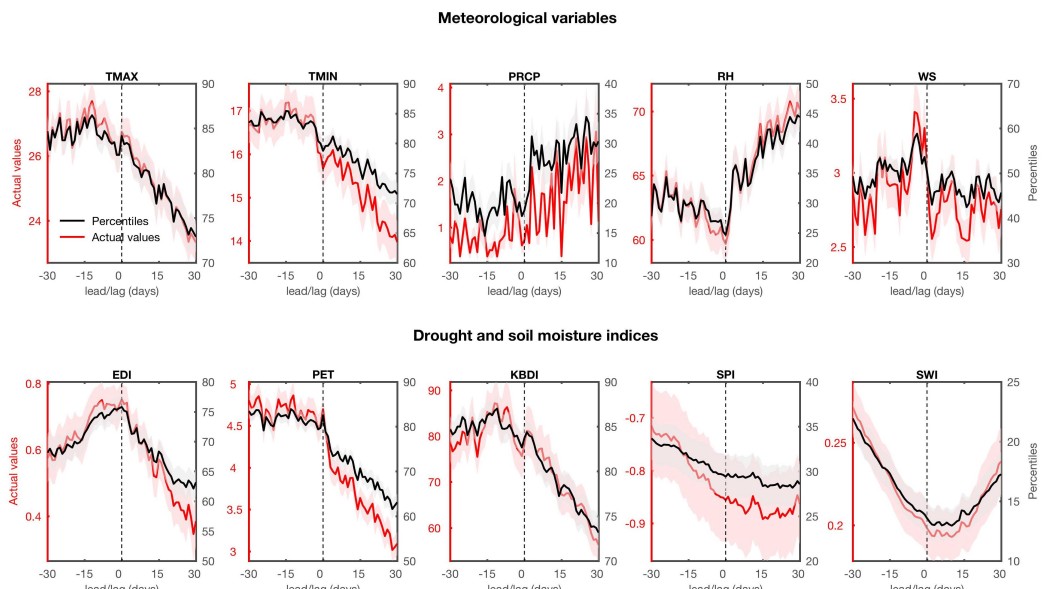

**Figure A2.** Same as Figure A1 but for meteorological, drought and soil moisture indices.



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
