# Peer review of "Simulating the effects of weather and climate on large wildfires in France"

_Natural Hazards and Earth System Sciences, 2018_

## Referee Comment (RC1) · Anonymous Referee #1 · 22 Nov 2018

General Comments:

The paper "Simulating the effects of weather and climate on large wildfires in France" makes a nice approach to model wildfire occurrence in France, using meteorological and fire danger indices. On the overall I think the paper is very well structured and written, so the message is quite clear. The methodology is well described and I don't find major issues regarding the publication of the work. I only have some minor issues, as described below.

Specific Comments:

Regarding Fig.2, a few suggestions: 1) Maybe presenting in each panel the total number of large fires (as presented in Table 3) would help the interpretation and context

of the plots; 2) Did the authors tried to look at these plots in log scale? (at least for Mediterranean Mountains region); 3) I find some of the statistically significant correlations between the number of large wildfire frequency and Total Burned Area confusing, with just a few cases, and most of them in the 0 class (the North region is a perfect example)

I wonder if there was the possibility of reducing the pool of predictors? By this, I mean looking for redundant/similar predictors amongst each group (Met.variable; Fire-Weather metric; Drought metric). Could it be the case that some of them provide very similar information, and consequently very similar performances for the models? In particularly, when bootstrap is lower and more combinations are possible, are the differences in the performance of these large enough to justify that there are no common predictors amongst models chosen for each area? The methodology is OK and well explained, but I wonder, if that was the case (not shown by the authors), wouldn't more "coherent" models in terms of more restricted predictors be more easily interpreted and also more coherent for further usage in other works and other realms?

Besides the prospect of using this scheme for future fire behavior, do the authors think it be usable/adapted on an Operational basis? I would like to see some discussion about this.

Other Comments:

Line 50: Could the authors introduce the meaning of the SAFRAN acronym in the text?

Line 19: "most extreme years"

Line 69: correct the title 2.4.1 "Generalized"

---

## Referee Comment (RC2) · Fox (Referee) · 21 Dec 2018

General comments: The manuscript analyses the usefulness of a range of weather related variables in predicting the occurrence of large wildfires (>100 ha). The main fire zone of France is divided into 6 regions and explanatory variables are tested using logistic regression.

The English is generally good but the paper could be improved with some restructuring. In its current state, some Results (fire data, regression equations) are presented in the Methods and much of the Discussion is in the Conclusions. All results of data treatment by authors should be moved out of Methods and into the Results. The Conclusions should summarize the main points of the Results & Discussion but not introduce new

information / interpretations.

Specific comments: The explanatory variables are a bit confusing. The first 6 Meteorological variables should be deleted from the study; many of these variables are used to calculate the indices/metrics below, so they're accounted for elsewhere (with potential problems of covariance), explaining perhaps why they're of no significance in any of the regressions. How the indices/metrics are calculated should be presented in the paper so that the weather variables used to calculate them are explicit for readers.

Fire data should be presented more extensively in the Results: Table 2 should include total number of fires and burned area per region, number of fires > 100 ha, contribution of fires >100 ha to RBA, contribution of fires >100 ha to NBA.

Explanatory variable characteristics related to fires >100 ha should be described in the Results section so readers working on large fires can relate thresholds to their own context. As it is, the regressions show whether variables are significant or not, but they give no indication of the range of explanatory values involved in large fires.

The absence of wind as an explanatory variable in most of the regions should be discussed more fully. Very large fires occur only in very windy conditions, so it's somewhat surprising that wind is significant only in Mdt North (FWI). Similarly, results of some of the regions suggest that fire-weather is insignificant in large fires and only the state of the vegetation or litter layer counts. This also could be discussed more fully, and significant indices / metrics should be related more explicitly to weather / climate in keeping with the title of the paper.

Technical corrections: A number of minor points / suggestions have been annotated in the manuscript, but these will be sent directly to the authors.
* * *

---

## Author Comment (AC1) · 12 Feb 2019

General Comments:

The paper "Simulating the effects of weather and climate on large wildfires in France" makes a nice approach to model wildfire occurrence in France, using meteorological and fire danger indices. On the overall I think the paper is very well structured and written, so the message is quite clear. The methodology is well described and I don't find major issues regarding the publication of the work. I only have some minor issues, as described below.

We thank the reviewer for their very positive comments to our manuscript.

Specific Comments:

Regarding Fig.2, a few suggestions: 1) Maybe presenting in each panel the total number of large fires (as presented in Table 3) would help the interpretation and context of the plots;

This is a good suggestion. We added in the total number of large fires in the upper-left corner of each panel (see below).

[Figure]

Figure 2. Interannual relationships between the annual frequency of large wildfires (>100 ha) and the total annual burned area. The total number of large wildfires as well as Pearson correlations are indicated for each region. The symbol * indicates significant correlations at the 95% confidence level. The linear fitting and the 95% confidence intervals are also shown.

2) Did the authors tried to look at these plots in log scale? (at least for Mediterranean Mountains region);

Some years in the Mediterranean Mountains regions have seen no wildfires (at least through the lens of MODIS) and the total burned area is equal to 0. The issue when using a log scale is that 0 is undefined. We thus prefer to stick with a linear scale.

3) I find some of the statistically significant correlations between the number of large wildfire frequency and Total Burned Area confusing, with just a few cases, and most of them in the 0 class (the North region is a perfect example)

The reviewer is probably referring to Table 2 (last column). We agree that the caption was unclear. This is actually showing the correlations between the annual frequency of large wildfires and the total annual burned area, as illustrated in Figure 2. For clarity, we moved this information to Figure 2 (see above). Note that the correlations are even higher when using the non-parametric Spearman Rank correlation method, as expected given the non-linear (monotonic) relationship between the annual number of large fires and the total annual burned area.

I wonder if there was the possibility of reducing the pool of predictors? By this, I mean looking for redundant/similar predictors amongst each group (Me variable; Fire- Weather metric; Drought metric). Could it be the case that some of them provide very similar information, and consequently very similar performances for the models? In particularly, when bootstrap is lower and more combinations are possible, are the differences in the performance of these large enough to justify that there are no common predictors amongst models chosen for each area? The methodology is OK and well explained, but I wonder, if that was the case (not shown by the authors), wouldn't more "coherent" models in terms of more restricted predictors be more easily interpreted and also more coherent for further usage in other works and other realms?

The reviewer raises a good point. We believe that both approaches are valuable. The advantage when using a more limited set of predictors is that models are expected to be more coherent from a region to another and more straightforward to run on the future period. However, sticking only with a few indices would raise another question: which indices are the most appropriate in France to track wildfire potential? Each fire weather index is reasoned to have different sensitivities to temperature, wind speed, relative humidity and precipitation, and selecting an index a priori would be subjective. This is the reason why we opted for a "let the data decides" approach. We believe that this approach is well suited for such an exploratory analysis and will serve to support the use of the CFFDRS metrics (and especially the FWI) as well as the SWI in further climate-fire studies in France.

However, meteorological variables were not selected in any ecoregion as significant predictor of large wildfire. This confirms previous findings that biophysical variables are doing a better job in tracking wildfire activity. Based on this finding and reviewer comment, we decided to delete the 6 meteorological variables from the study, thereby reducing the pool to 14 predictors.

Here is the new pool of predictors used in the modelling framework:

| Name | Acronym | Category |
| --- | --- | --- |
| 1. Fine Fuel Moisture Code | FFMC | Fire-Weather metric |
| 2. Duff Moisture Code | DMC | Fire-Weather metric |
| 3. Drought Code | DC | Fire-Weather metric |
| 4. Initial Spread Index | ISI | Fire-Weather metric |
| 5. Build-Up Index | BUI | Fire-Weather metric |
| 6. Fire Weather Index | FWI | Fire-Weather metric |
| 7. Forest McArthur Fire Danger Index | FFDI | Fire-Weather metric |
| 8. F-Index | FINDEX | Fire-Weather metric |
| 9. Nesterov Fire Danger Index | NFDI | Fire-Weather metric |
| 10. Fosberg Fire Weather Index | FFWI | Fire-Weather metric |
| 11. Effective drought Index | EDI | Drought metric |
| 12. Potential Evapotranspiration | PET | Drought metric |
| 13. Standardized Precipitation Index | SPI | Drought metric |
| 14. Soil Wetness Index | SWI | Soil Moisture metric |

Besides the prospect of using this scheme for future fire behavior, do the authors think it be usable/adapted on an Operational basis? I would like to see some discussion about this.

Indeed, the idea behind this modelling framework is to provide a basis to simulate future changes to wildfires but the reviewer raises a good point. This model could be used in a real-time fashion. We added in the following sentences in the Results and Discussion section:

"This modelling framework has multiple potential applications. First, it could be implemented in a real-time fashion using meteorological forecasts. This may complement traditional forecasts based on FWI only. Indeed, the FWI only measures the potential intensity of wildfire and this quantity is not always straightforward in the real world. In this regard, our model translates a series of fire weather and drought indices into a probability of occurrence of large wildfire that could be useful in decision-making".

Other Comments:
Line 50: Could the authors introduce the meaning of the SAFRAN acronym in the text?

Done. SAFRAN stands for "Système d'Analyse Fournissant des Renseignements Atmosphériques a la Neige" (Analysis system providing data for snow model).

Line 19: "most extreme years"

Good catch. We corrected.

Line 69: correct the title 2.4.1 "Generalized"

We corrected. Many thanks for this review.

---

## Author Comment (AC2) · 12 Feb 2019

General comments: The manuscript analyses the usefulness of a range of weather related variables in predicting the occurrence of large wildfires (>100 ha). The main fire zone of France is divided into 6 regions and explanatory variables are tested using logistic regression.

The English is generally good but the paper could be improved with some restructuring. In its current state, some Results (fire data, regression equations) are presented in the Methods and much of the Discussion is in the Conclusions. All results of data treatment by authors should be moved out of Methods and into the Results. The Conclusions should summarize the main points of the Results & Discussion but not introduce new information / interpretations.

We thank the reviewer for their positive comments to our manuscript. We followed reviewer's suggestion and restructured some parts of the manuscript to improve readability.

Specific comments: The explanatory variables are a bit confusing. The first 6 Meteorological variables should be deleted from the study; many of these variables are used to calculate the indices/metrics below, so they're accounted for elsewhere (with potential problems of covariance), explaining perhaps why they're of no significance in any of the regressions.

The reviewer raises a good point. We included meteorological variables with the intention of detecting specific processes such as heat wave or wind spells pertaining to wildfire (these processes being mixed in fire weather indices). However, meteorological variables were not selected in any ecoregion as significant predictor of large wildfire. This confirms previous findings that biophysical variables are doing a better job in tracking wildfire activity. Based on this finding and reviewer comment, we decided to delete the 6 meteorological variables from the study, thereby reducing the pool to 14 predictors (see below).

| Name | Acronym | Category |
| --- | --- | --- |
| 1. Fine Fuel Moisture Code | FFMC | Fire-Weather metric |
| 2. Duff Moisture Code | DMC | Fire-Weather metric |
| 3. Drought Code | DC | Fire-Weather metric |
| 4. Initial Spread Index | ISI | Fire-Weather metric |
| 5. Build-Up Index | BUI | Fire-Weather metric |
| 6. Fire Weather Index | FWI | Fire-Weather metric |
| 7. Forest McArthur Fire Danger Index | FFDI | Fire-Weather metric |
| 8. F-Index | FINDEX | Fire-Weather metric |
| 9. Nesterov Fire Danger Index | NFDI | Fire-Weather metric |
| 10. Fosberg Fire Weather Index | FFWI | Fire-Weather metric |
| 11. Effective drought Index | EDI | Drought metric |
| 12. Potential Evapotranspiration | PET | Drought metric |
| 13. Standardized Precipitation Index | SPI | Drought metric |
| 14. Soil Wetness Index | SWI | Soil Moisture metric |

How the indices/metrics are calculated should be presented in the paper so that the weather variables used to calculate them are explicit for readers.

More information on how these indices are computed are already available in the literature. A specific reference is provided for each index in the current version of the manuscript. We feel like adding more details on each index would considerably slow down the paper.

Fire data should be presented more extensively in the Results: Table 2 should include total number of fires and burned area per region, number of fires > 100 ha, contribution of fires >100 ha to RBA, contribution of fires >100 ha to NBA.

Good suggestion. We added in this information in Table 2 (see below). Also, for clarity we moved the correlation between the annual frequency of large wildfires and total burned area to Figure 2.

**Table 2.** This table provides for each environmental region (first column): the number of wildfires (second column), the number of large wildfires (>100 ha) (third column), the contribution of large wildfires to regional burned area (fourth column) and the contribution of large wildfires to national burned area (fifth column).

| Env. Region | # Wildfires | # Large wildfires | Contribution to regional BA | Contribution to national BA |
|---|---|---|---|---|
| North | 49 | 6 | 61.8% | 1.9% |
| Alpine | 41 | 8 | 69.1% | 2.3% |
| West | 101 | 19 | 63.6% | 5.4% |
| Mediterranean Mountains | 289 | 51 | 83.9% | 34.8% |
| Mediterranean North | 309 | 59 | 75.9% | 25.7% |
| Mediterranean South | 105 | 13 | 72.4% | 7.1% |

Explanatory variable characteristics related to fires >100 ha should be described in the Results section so readers working on large fires can relate thresholds to their own context. As it is, the regressions show whether variables are significant or not, but they give no indication of the range of explanatory values involved in large fires.

This is a good point. We added in a new table (see below) in the supplementary information illustrating the range (95% confidence interval) of each significant predictor variable during the day of large wildfires for each region. This gives an overall idea of the typical conditions during which large wildfires occurred during the studied period.

**Table A1.** Typical range of explanatory variables during the day of large wildfires. The range indicates the 2.5 and 97.5 percentile (95% confidence interval) of the composite means obtained from 1,000 bootstrapped datasets.

| Env. Region | Predictor 1 (95%CI) | Predictor 2 (95%CI) | Predictor 3 (95%CI) |
|---|---|---|---|
| North | FFMC(76.7;82.5) | | |
| Alpine | SPI(-2.0;-1.1) | | |
| West | DC(661.6;767.4) | | |
| Mediterranean Mountains | DMC (86.1;108.9) | SWI(0.14;0.18) | |
| Mediterranean North | FWI(26.9;30.9) | ISI(7.0;8.3) | SWI(0.12;0.14) |
| Mediterranean South | DMC(77.2;131.2) | | |

The absence of wind as an explanatory variable in most of the regions should be discussed more fully. Very large fires occur only in very windy conditions, so it's somewhat surprising that wind is significant only in Mdt North (FWI). Similarly, results of some of the regions suggest that fire-weather is insignificant in large fires and only the state of the vegetation or litter layer counts. This also could be discussed more fully, and significant indices / metrics should be related more explicitly to weather / climate in keeping with the title of the paper.

We discussed more deeply the absence of wind speed as significant predictor as well as the lack of fire weather signal in some ecoregions:

« […] It is noteworthy that the effect of wind speed on large wildfires is only revealed through the ISI in the Mediterranean North. The absence of wind speed as a significant factor in other regions may arise due to the temperature decrease associated with wind spells in the French Mediterranean (Ruffault et al., 2017b), with contrasting effects on commonly used fire weather indices that were designed to increase with temperature. This may also indicate the stronger role of fuel moisture in these regions in response to slower climatic variations, regardless what short-term fire weather does. »

Technical corrections: A number of minor points / suggestions have been annotated in the manuscript, but these will be sent directly to the authors.

We thank the reviewer for their suggestions that helped improve the manuscript.